# Fermented Minor Grain Foods: Classification, Functional Components, and Probiotic Potential

**DOI:** 10.3390/foods11203155

**Published:** 2022-10-11

**Authors:** Huibin Qin, Houbin Wu, Ke Shen, Yilin Liu, Meng Li, Haigang Wang, Zhijun Qiao, Zhixin Mu

**Affiliations:** 1Center for Agricultural Genetic Resources Research, Shanxi Agricultural University, Key Laboratory of Crop Gene Resources and Germplasm Enhancement on Loess Plateau, Ministry of Agriculture, Shanxi Key Laboratory of Genetic Resources and Genetic Improvement of Minor Crops, Taiyuan 030031, China; 2Shennong Technology Group Co., Ltd., Jinzhong 030801, China

**Keywords:** minor grain, fermented food, classification, functional components, probiotic potential

## Abstract

Fermented minor grain (MG) foods often have unique nutritional value and functional characteristics, which are important for developing dietary culture worldwide. As a kind of special raw material in fermented food, minor grains have special functional components, such as trace elements, dietary fiber, and polyphenols. Fermented MG foods have excellent nutrients, phytochemicals, and bioactive compounds and are consumed as a rich source of probiotic microbes. Thus, the purpose of this review is to introduce the latest progress in research related to the fermentation products of MGs. Specific discussion is focused on the classification of fermented MG foods and their nutritional and health implications, including studies of microbial diversity, functional components, and probiotic potential. Furthermore, this review discusses how mixed fermentation of grain mixtures is a better method for developing new functional foods to increase the nutritional value of meals based on cereals and legumes in terms of dietary protein and micronutrients.

## 1. Introduction

Minor grains (MGs) comprise the majority of cereal crops, except for rice and wheat, which are categorized as coarse cereals and legumes [1]. In general, coarse cereals include maize (*Zea mays*); sorghum (*Sorghum vulgare*); oats (*Avena sativa*); barley (*Hordeum vulgare*); buckwheat (*Fagopyrum esculentum*); quinoa (*Chenopodium quinoa*); and some millet species, including pearl millet (*Pennisetum glaucum*), finger millet (*Eleusine coracana*), foxtail millet (*Setaria italica*), and kodo millet (*Paspalum scrobiculatum*) [2]. Legumes (e.g., Bengal gram, broad beans, black gram, mung beans, lentils, and peas) are also considered MGs, because their dietary nutrient properties are similar to those of coarse cereals [3]. MGs are mainly grown in marginal areas and usually exhibit strong environmental and climate adaptability, such as drought resistance and cold- and alkali resistance [4]. In general, MG grains are nutritionally superior compared to staple grains, as MGs are richer in nutritionally valuable components that include fibers, vitamins, and minerals (especially trace elements such as iron and zinc) [5]. MGs are required for sustaining human health and enhancing dietary composition, since they are abundant in useful components [6]. Polyphenols, β-glucans, alkaloids, and a small quantity of anthraquinones are the main functional components of GMs, which regulate blood glucose, blood lipids, and blood pressure levels; improve hepatic non-alcoholic fatty activity; and lower the risk of cardio-cerebrovascular disease [7,8,9]. Nutritional deficiencies of MGs, however, include the absence of some nutrients, such as proteins and some essential amino acids (lysine), low starch utilization, and the possible presence of some anti-trophic factors (phytic acid, tannins, phenolic acid, etc.) [10].

Fermented foods and beverages were redefined in 2019 by the International Scientific Association for Probiotics and Prebiotics (ISAPP) as foods made through desired microbial growth and enzymatic conversions of food components [11]. Fermented MG foods refer to the fermentation of coarse cereals and legumes other than rice and wheat by microorganisms on the surface of minor grains or artificially inoculated beneficial microorganisms used to change the grain’s nutritional composition and produce a unique flavor. Compared to foods cooked with raw grain ingredients, fermented MG foods not only increase the shelf life of foods but are also generally more enjoyable and digestible and contain various nutrients, such as free amino acids, organic acids, and vitamins [12]. Fermented foods have been recognized for their unique role in human history, with various fermented products becoming a part of cooking and a worldwide cultural heritage in many countries worldwide. In addition to the importance of fermented foods for public health and food preservation and quality, current epidemiological evidence shows that diets high in fermented foods can lower disease risk and improve health and quality of life [13].

Fermented MG foods come from a wide range of sources and are simple to manage, and fermented products can inhibit the growth of most pathogens [14]. Locally sourced fermented MG foods are very popular in many developing countries [15]. Among the existing fermented products, with the enhancement of consumer health awareness, fermented products based on MGs as raw materials have once again become favored by consumers [16]. Traditional fermented MG products are often consumed regionally. Although not as prevalent as other fermented foods, fermented MGs often represent local culture and traditions [17]. For example, cereals such as sorghum and maize porridge have been a staple in Africa and South America. Many fermented and unfermented sorghum, millets, wild legume seeds, meat, and dairy products are used as main foods [18,19]. For Africans, traditional MG foods are essential for storing inexpensive meals, enhancing the digestibility and nutritional value of components, and fermenting to enhance the flavor of existing grains [20].

The microorganisms that cause fermentation may have originated on the substrate or been introduced as a starter [21]. Some fermented foods have a richer nutritional content than others, and consumers can gain health advantages from this because of the microbial conversion of the chemical composition of the raw materials during fermentation [19]. Through various natural fermentation processes, lactic acid bacteria (LAB), yeasts, and/or fungi ferment various substrates of coarse cereals (often combined with other legumes) to produce various types of food [22]. Through microbial action during fermentation, proteins and carbohydrates create bioactive compounds [23]. Metabolite compounds play an important role in preventing chronic diseases such as obesity, diabetes, cardiovascular disease, cancer, and allergies [24,25].

Complex phytochemical compounds degrade during fermentation into smaller, more bioactive polyphenols. Studies have demonstrated that polyphenols present in fermentation products are advantageous for the growth and metabolism of the microbiome [26], as well as for reducing inflammatory reactions and cytokine production [27]. Additionally, reducing oxidative stress, reducing free radicals, controlling antioxidant enzyme activity, increasing immune system activity, and neutralizing free radicals are all plausible mechanisms for the beneficial effects of fermented MG foods and beverages. The impact of fermented MG products (such as sour bread, porridges, kefir, beers, wines, and vinegar) generated or preserved by microorganisms on general health, specifically their significance on the balance of the gut microbiota and brain functionality, is now well established [16].

Nevertheless, apart from yogurt and other dairy products, there are few well-designed experiments that have investigated the health benefits of various fermented foods [28,29]. Similarly, hypothesis-driven studies that describe how fermented MG foods affect human physiological mechanisms are limited [30]. Many local or traditional fermented MG foods are still prepared by hand in households, villages, and small industries. In contrast, the production of vinegar, soy sauce, and various alcoholic beverages has developed to a biotechnological level and been widely commercialized [31]. The number of articles and books on locally produced fermented foods and beverages has expanded during the past twenty years. Recently, fermentation technology has received great renewed attention because fermentation can provide a solid guarantee for the unique nutrition and function of safe food [32].

The goal of this review is to enumerate and analyze some of the most common traditional fermented MG foods, classify them according to the fermentation process and fermented product types, analyze changes in cereal nutrients after fermentation, and explore the probiotic potential of fermented MG foods. Currently, people are increasingly demanding food that is convenient, nutritious, and safe. Therefore, it is very important to elucidate the microbial diversity and nutritional components of traditional fermented MG foods, make the original production process more convenient, and improve the industrialization level of traditional fermented food. Furthermore, with the development of modern biotechnology, there will be more new technologies and new processes to solve the problems in product traits, flavor substances, and preparation of traditional fermented MG foods. At the same time, exploring the functional factors and probiotics is beneficial for developing the fermented MG food industry.

## 2. Classification of Fermented MG Foods

In many countries, traditional fermented foods represent the historical and cultural heritage of local foods, as they enhance the sensory properties of raw materials and have a unique ability to preserve developed products. It is well known that microorganisms (either naturally present or intentionally added) determine the fermentation processes and outcomes as well as the characteristic properties of the final fermented foods [11]. Four main fermentation processes are present in the dominant metabolic process: lactic, alkaline, acetic, and alcoholic fermentation [12,21,28]:Lactic fermentation is mainly carried out by LAB and different types of organic acids, of which lactic acid is the main metabolite.Alkali fermentation is carried out by different species of *Bacillus* and fungi and often occurs during seed fermentation.Acetic fermentation is mainly carried out by actic acid bacteria (AAB). In the presence of excess oxygen, AAB (mainly *Acetobacter*) convert alcohol to acetic acid, with acetic acid as the primary product.Alcohol fermentation, with yeasts as the predominant organisms, results in the primary products of ethanol and CO_2_.

Almost all MGs can be made into different kinds of food by using different natural fermentation methods. A range of locally fermented foods worldwide made from MGs is listed in Table 1, Table 2, Table 3 and Table 4. It can be observed from these tables that most of the products made using fermented MGs (minor grains or minor grains in combination with legumes) are found in Africa and Asia. Moreover, adding legumes improves the overall protein quality of fermented foods [33]. Certain fermented MG foods do not ferment on their own and require adding some of the major grains, resulting in more balanced and even nutrition for fermented foods.

A detailed understanding of microorganism diversity and the fermentation processes of traditional fermented foods is a prerequisite for investigating how to improve the nutritional and safe quality of these foods. Some of the fermented MG foods, classified by metabolic process and fermented product types, are described in the following sections.

### 2.1. Lactic Acid-Fermented MG Pancakes and Bread

The typical products of lactic fermentation are sour pancakes and bread, mainly caused by LAB [34]. LAB species found in different traditional fermented MG foods include *Enterococcus*, *Lactobacillus*, *Lactococcus*, *Leuconostoc*, *Oenococcus* and *Pediococcus*, *Streptococcus*, *Tetragenococcus*, *Vagococcus,* and *Weissella* [35,36,37]. During lactic fermentation, through acidification and fermentation, LAB produce organic acids, primarily lactic acid and acetic acid, which are the most effective antimicrobial agents in reducing the pH of food below 4.0, thus preventing the survival of harmful microorganisms [38,39]. In addition to producing organic acids, LAB can also produce hydrogen peroxide by oxidizing reductive nicotinamide adenine dinucleotide flavin nucleotides, an enzyme that reacts rapidly with oxygen [40]. Due to the absence of real catalase, LAB can build up and hinder some microbes by failing to degrade the hydrogen peroxide created [41].

During the lactic fermentation process, endogenous grain amylase produces fermentable sugar as an energy source for LAB, and some kinds of LAB can degrade starch [42,43]. In addition, LAB help maintain a healthy composition of intestinal flora and improve local and systemic immunity [44]. One of the most widely used bacteria for commercial purposes today, *Lactobacillus,* is frequently found in yogurt, kimchi, and other foods. It is the most common microorganism in human nutrition and fermented food systems [45,46,47].

Table 1 lists several varieties of sour pancakes and bread from different regions. The quality and flavor of products are enhanced by the lactic fermentation of bread dough. Additionally, the fermentation improves the flavor of bread made with inferior flours and underutilized MGs [48]. Lactic-fermented MG pancakes and breads are still a significant staple food for people in Africa and several regions of Asia. Some examples of similar fermentation products are India’s Adai, Dhokla, Dosa, Idli, Ethiopia’s Enjera/Injera, and Sudan’s Hussuwa.

Idli, a traditional fermented bread in India, is a pastry made with rice and black gram flour that has a sourdough bread-like texture. Idli is a mixture of rice and dehulled black gram ingredients that are wet-ground and fermented traditionally. It is recognized for its delicate texture, scrumptious acidity, and distinctive aroma [10,49].

Injera (Enjera) is the indisputable national food of Ethiopia. It can be produced using a variety of grains, such as sorghum, corn, millet, tef, and barley. To make injera, grains are shelled and ground into flour. After water is added to make a dough, a leavening agent (double) is added, and the dough is then fermented for 2–3 days. A sourdough kept from the previously fermented dough is the starter. After fermentation, the dough is thinned to a thick batter and placed onto a pan that has been lightly oiled. The pan is then tightly covered with a lid to keep the steam in. The injera emerges from the pan and is prepared for serving in about two minutes. The storage term at standard temperature is no longer than three days. Standard injera has circular, uniformly distributed honeycomb “eyes” on top and is soft, elastic, and round. The flavor of premium injera is mostly distinguished by its somewhat acidic flavor [21].

Hussuwa is a fermented steamed cake, similar to dough, created in Sudan from sorghum or millet malt and a substance known as “aceda” (stiff porridge made from unmalted flour). Hussuwa is a thick paste made from sorghum flour and water combined in equal parts that is cooked to create the firm “aceda” porridge. A second half-volume of sorghum malt is then added and left to ferment for up to 48 h to produce ajin, a type of sourdough that is baked on a hot plate until all moisture has been removed. It was determined that during mashing and acidification, *Lactobacillus fermentum* dominated the LAB species present in the malt [50].

**Table 1 foods-11-03155-t001:** Examples of lactic acid-fermented MG pancakes and bread used in different regions of the world.

Name	Grain	Microorganism (s)	Country/Region	Type of Product	Reference
Adai	Kodo millet, barnyard millet, and legumes	*Pediococcus,* *Streptococcus,* *Leuconostoc.*	India, Sri Lankan	Steamed cake	[21]
Dhokla	Rice or wheat and bengal gram	*Leuconostoc,* *Lactobacillus,* *Streptococcus,* *Torulopsis.*	Northern India	Steamed cake	[19]
Dosa	Rice and bengal gram	*Leuconostoc,* *Lactobacillus,* *Streptococcus,* *Torulopsis,* Yeasts.	India	Griddled cake	[21]
Enjera/Injera	Sorghum, tef, barley, finger millet, maize, or wheat	*Leuconostoc,* *Saccharomyces,* *Lactobacillus.*	Ethiopia, Africa	Pancake-like bread	[21,35]
Hussuwa	Sorghum	*Lactobacillus,* *Pediococcus,* Yeasts.	Sudan	Steamed cake	[19]
Idli	Rice grits and black gram	*Leuconostoc,* *Streptococcus,* *Torulopsis,* *Candida,* *Tricholsporon.*	South India, Sri Lanka	Steamed cake	[19]
Kisra	Sorghum	*Lactobacillus,* *Candida.,* *Debaryomyces,* *Saccharomyces.*	Sudan	Pancake-like bread	[21,51]
Kwunu-Zaki	Millet, sorghum, or maize	LAB, Yeasts.	Northern Nigeria	Paste	[21]
Mungbean starch	Mungbean	*Leuconostoc,* *Lactobacillus.*	China, Thailand, Korea, Japan	Noodle	[35]
Rye bread	Rye	LAB	Denmark	Sandwich bread	[35]
Tempeh	Whole-grain barley and oat	*Rhizopus.*	Indonesia	Cake	[33]
Ogi, Ogi-Baba	Maize, millet, and sorghum	*Lactobacillus.*	Nigeria, Westen Africa	Pudding	[12]

### 2.2. Lactic- and Acetic-Fermented MGs and Non-Alcoholic Porridge and Beverages

Ting from southern Africa, Ben-saalga from Burkina Faso, Boza from Bulgarian, Ogi from Nigeria, and Suanzhou from China are a few examples of sour porridge. Fermented ingredients, such as maize, millet, sorghum, or cassava, are then wet-milled, wet-sieved, and boiled to create the sour porridge.

In some of these fermented porridges and beverages, yeast is also present and may produce an amount of alcohol, but the number of yeasts is not dominant. Yeast growth is inhibited by the growth of LAB or AAB, resulting in a low alcohol concentration of less than one percent in the finished product [52]. The fermentation time of acid-fermented foods is often short, 1–3 days, and after fermentation, the pH is low, ranging from 3.5 to 4.4. Although yeasts do not seem to have much of an effect on fermentation, the flavor and preservation of the finished product may be impacted by their higher numbers in the last stage of the fermentation process. Species of *Saccharomyces*, *Picha,* and *Candida* are capable of proliferating in sour porridge under low pH [53].

Ting is a locally produced, fermented, sorghum-based porridge that is popular in South Africa and its neighboring nations. Mycotoxin levels in the ting samples are dramatically reduced during fermentation, especially when LAB strains are used. It is widely eaten by adults and frequently used to feed babies [54].

Ben-saalga is a well-liked, fermented gruel made in Burkina Faso from pearl millet. Pearl millet is typically processed into Ben-saalga using the following steps: washing (optional), soaking the grains (first fermentation step), grinding and sieving the wet flour, settling (second fermentation step), and cooking. Though lactic acid fermentation takes place during the settling step, alcoholic fermentation predominates during the soaking step. A temporal fluctuation of metabolic activity is indicated by fermentation kinetics during settling. Only a homofermentative pathway is active at first, then both homofermentative and heterofermentative pathways became active. The paste that was created after settling had a low pH (4.0 ± 0.4), and LAB predominated the microflora [55].

Boza is a traditional fermented beverage with a nice, sweet-sour, bread-like flavor that is popular in Bulgaria. Some regions of Romania, Albania, and Turkey also consume this drink. Boza can be made from a variety of grains, including wheat, millet, and rye, and fermentation is brought on by organic yeast and LAB mixes. Microorganism interactions are not under control during the procedure, which leads to variability in product quality [56].

Ogi is a fermented cereal gruel. It has a smooth, porridge-like texture, a tart flavor like yoghurt, and a distinct aroma. Ogi’s color is said to vary depending on the type of cereal grain used to make it: cream-white for maize, reddish-brown for sorghum, and dirty grey for millet. It is consumed throughout Nigeria as a weaning meal. The traditional way to make Ogi in Nigeria is by steeping the grains of interest in earthenware, plastic, or enamel pots for 1–3 days to allow them to ferment. The Ogi slurry is produced by wet-milling of fermented grains. After wet-screening, there may be an optional further fermentation for one to three days in some communities [57].

Suanzhou is a traditional northwestern fermented food produced in China within agrarian communities using millet proso, millet foxtail, glutinous rice and other kinds of cereals. The preceding fermentation’s supernatant or fermentation soup is used to soak the grains, which are then left at room temperature for 24 h in a lidded jar. Grain that has been fermented is removed for cooking. To the acidic soup, raw materials are once again added and steeped in preparation for fermentation and consumption. In the fermentation of Suanzhou, LAB, AAB, and yeast are the three main species involved [58].

**Table 2 foods-11-03155-t002:** Examples of lactic- and acetic-fermented MGs and non-alcoholic porridge and beverages used in different regions of the world.

Name	Grain	Microorganism (s)	Country/Region	Type of Product	Reference
Ambali	Millet	*Leuconostoc,* *Lactobacillus,* *Streptococcus.*	India	Porridge	[42]
Bagni	Millet	*Lactobacillus.*	Caucasus	Liquid drink	[21,42]
Ben-saalga	Pearl millet	*Lactobacillus,* *Pediococcus,* *Leuconostoc,* *Weissela,* Yeasts.	Burkina Faso, Ghana	Weaning food, beverage	[55]
Bogobe	Sorghum	Unknown	Botswana	Soft porridge	[19]
Boza	Millet, wheat, rice, maize, barley, oat, or rye	*Leuconostoc,* *Lactobacillus,* *Saccharomyces,* *Candida,* *Candida,* *Geotrichum.*	Bulgaria, Romania, Albania, Turkey, Republic of Northern Macedonia, Romania, Southern Russia, Northern Africa	Sour refreshing liquid	[19]
Braga	Millet	Unknown	Romania	Liquid drink	[19]
Busa	Maize, millet, or sorghum	*Saccharomyces,* *Schizosacchromyces,* *Lactobacillus,* *Leuconostoc,* *Pediococcus.*	Syria, Egypt, Kenya, Turkey, eastern Africa	Liquid drink	[21,35]
Bussa	Maize, sorghum, or malt	*Lactobacillus,* *Saccharomyces,* *Penicillium,* *Candida.*	Kenya, Nigeria, Ghana	Food, refreshment drink	[21,35]
Bushera (Obushera)	Millet or sorghum flour	*Lactobacillus,* *Streptococcus.*	Western Uganda	Beverage	[12]
Burukutu	Sorghum and cassava	LAB, *Candida,* *Saccharomyces.*	Nigeria	Creamy drink with suspended solids	[10]
Dalaki	Sorghum	Unknown	Nigeria	Thick porridge	[19]
Darassum	Millet	Unknown	Mongolia	Liquid drink	[19]
Hulumur	Red sorghum	*Lactobacillus.*	Sudan	Clear drink	[35]
Kunu-zaki	Maize soghum, or millet	*Lactobacillus,* *Cephalosporium,* *Aerobacter,* *Candida,* *Saccharomyces,* *Rhodotorula,* *Corynebacterium,* *Fusarium,* *Aspergillus,* *Penicillium.*	Northern Nigeria	Acidic porridge	[19,42]
Koko	Millet	*Lactobacillus,* *Weissella.*	Northern Ghana	Beverage	[19]
Koozh	Millet, rice	*Lactobacillus.*	Southern India	Beverage	[12]
Kunu	Maize, millet, or sorghum	*Lactobacillus,* *Leuconostoc,* *Pediococcus,* *Weissella.*	Western Africa	Beverage	[12]
Kvass	Rye and barley malt	*Lactobacillus,* *Saccharomyces.*	Central and Eastern Europe	Beverage	[12]
Kwunu-Zaki	Millet, sorghum, or maize	LAB	Northern Nigeria	Beverage	[12]
Mahewu	Millet, sorghum, maize, or wheat flour	*Lactobacillus,* *Streptococcus.*	Zimbabwe, Southern Africa	Sweet-sour, non-alcoholic drink, 8–10% dry matter DM clear drink	[21,35]
Nasha	Sorghum	*Streptococcus,* *Lactobacillus,* *Candida,* *Saccharomyces.*	Sudan	Porridge as a snack	[21]
Ogi-Baba/ ogi-gero	Sorghum, millet	*Lactobacillus,* *Candida,* *Corynebacterium,* *Cephalosporium,* *Aerobacter,* *Rhodotorula,* *Fusarium,* *Aspergillus,* *Penicillium.*	Nigeria, Western Africa	Sour porridge, main meal, weaning food for babies	[21,35]
Tarhana, Trahana	Wheat, rye, maize, barley, corn, oat, or buckwheat	*Lactobacillus,* *Streptococcus,* *Saccharomyces.*	Greece, Cyprus, Turkey	Soup	[12]
Thobwa	Sorghum	LAB	Malawi	Thin porridge drink	[12]
Thumba	Millet	*Endomycopsin.*	Eastern India	Liquid drink	[21]
Ting	Sorghum	*Lactobacillus,* *Weissella.*	Botswana and South Africa	porridge	[54]
Togwa	Cassava, maize, sorghum, or millet	*Lactobacillus,* *Pediococcus,* *Weissella,* *Candida,* *Issatchenkia,* *Saccharomyces.*	Tanzania	Fermented gruel or beverage for refreshment and weaning	[19]
Uji	Maize, millet, sorghum, or cassava flour	*Leuconostoc,* *Lactobacillus,* *Pediococcus.*	Kenya, Uganda, Tanzania	Sour porridge, main meal	[21]
Vada	Cereal and legume	*Pediococcus,* *Streptococcus,* *Leuconostoc.*	India	Breakfast or snack food	[21]
Suanzhou	Cereal	*Lactobacillus,* *Acetobacter,* *Pichia.*	Northwestern of China	Breakfast or snack food	[58]
Mbege	Maize, sorghum, or millet	*Saccharomyces,* *Schizosaccharomyces,* *Lactobacillus,* *Leuconostoc.*	Tanzania	Food, refreshment drink	[19,35]

### 2.3. Alcoholic and Lactic-Fermented Sweet/Sour Alcoholic MG Porridges

Some people consume fermented sweet/sour alcoholic MG porridges as their primary source of nutrition. Examples of alcoholic porridge or beverage involving acid fermentation are shown in Table 3. Asian examples include Pito, Burukutu, Munkoyo, Tella, and Sorghum beer.

Pito is a cream-colored alcoholic beverage, but “Burukutu” is a brown suspension. Both are brewed simultaneously using a single type of malted or germinated cereal grain or a combination of grains. The process goes as follows: soak the grains for a day, drain the soak water, let the seeds germinate for two days, let them dry in the sun, mill the grains into flour, mix the flour and water, boil the mixture for three to four hours to create slurry, settle and decant, add fresh water, and then reheat for three hours. The mixture is left to stand for 24 h at room temperature. The mixture is warmed for three hours while more water is added. Separation and cooling ensue. One day of fermentation at room temperature is used to produce Pito (the top clear supernatant) and burukutu from a supernatant and sediment (a thick brown suspension). Mycotoxin levels in Pito have been lowered by traditional fermenting techniques by 99% [59].

Munkoyo is a fermented sour grain drink with a low alcohol content that is popular in Congo and Zambia. Munkoyo is made by soaking corn and other grains, then saccharifying and liquefying them. Munkoyo broth is used in each batch of fermentation. In Munkoyo, lactic acid fermentation and alcohol fermentation are handled by *Lactobacillus confusus* and *Saccharomyces Cerevisiae*, respectively [60].

**Table 3 foods-11-03155-t003:** Examples of sweet/sour alcoholic fermented MG porridge used in different regions of the world.

Name	Grain	Microorganism (s)	Country/Region	Type of Product	Reference
Pito	Maize, sorghum, and sorghum	*Geotrichum,* *Lactobacillus,* *Candida.*	Nigeria, Ghana	Alcoholic cream-colored drink	[19]
Burukutu	Sorghum	*Saccharomyces,* *Leuconostoc,* *Candida,* *Acetobacter.*	Nigeria, Benin, Ghana	Alcoholic beverage of brown-colored suspension	[21]
Munkoyo	Corn, sorghum, millet, or maize plus munkoyo roots	*Lactobacillus,* *Saccharomyces.*	Katanga province, in southern Democratic Republic of Congo (D.R.C) and Zambia	Liquefied porridge	[21,60]
Sorghum beer	Sorghum, maize	LAB, Yeasts.	South Africa	Acidic, weakly alcoholic drink	[19]
Talla	Sorghum	*Unknown*	Ethiopia	Alcoholic drink	[19]
Chikokivana	Maize and millet	*Saccharomyces.*	Ziombabwe	Alcoholic beverage	[21]
Doro	Finger millet malt	Yeasts and bacteria.	Zimbabwe	Colloidal thick alcoholic drink	[21]
Jaanr	Millet	*Hansenula,* *Mucor.*	India, Himalaya	Alcoholic paste mixed with water	[21]
Kaffir	Malt of sorghum, maize	*Unknow*	Southern Africa	Beer	[35]
Merissa	Sorghum and millet	*Saccharomyces.*	Sudan	Alcoholic drink, bantu beer	[21,42]
Otika	Sorghum	*Unknown*	Nigeria	Alcoholic beverage	[21]

The traditional fermented alcoholic beverage known as tella is the most popular in Ethiopia. In the regions of Oromia, Amhara, and Tigray, it is the most widely consumed beverage. The materials used to make Tella include barley, wheat, maize, millet, sorghum, teff, and gesho leaves, as well as naturally occurring microbes. Although there is a lot of production and consumption, the fermentation process is still unexpected, spontaneous, and uncontrolled [61].

When making traditional sorghum beer, LAB are crucial in controlling the growth of putrid and hazardous microbes during the earliest stages of fermentation. Nevertheless, the alcohol produced by the yeast quickly stops the growth of the LAB [10].

### 2.4. Alkali-Fermented MG Food and Seasoning

There are much literature describing the nutrition of soy-fermented foods. Common soy-fermented foods are tempeh, douche, natto, miso, soy paste, soy sauce and dajang [62,63]. In this paper, we pay attention not only to soybean but to the fermented legumes and their nutritional characteristics.

*Bacillus* can be found in alkaline-fermented MG (mainly legumes) foods from Asia and Africa [64]. The main kinds of *Bacillus* in fermented MG foods include: *Bacillus amyloliquefaciens,*
*Bacillus*
*circulans, Bacillus coagulans, Bacillus firmus, Bacillus licheniformis, Bacillus*
*megaterium, Bacillus pumilus, Bacillus subtilis, Bacillus subtilis Variety Natto,* and *Bacillus*
*thuringiensis*. Some *Bacillus* subtilis, rich in λ-polyglutamate (PGA), is commonly found in fermented legumes foods in Asia, giving the products sticky characteristics [65].

Fermentation has improved legume shelf life, taste, appearance, nutrient digestibility, nutritional value, and texture, and protease inhibitors, lectins, oligosaccharides, and phytates (non-nutritional compounds) have been found to be reduced in legume seeds. Furthermore, legume fermentation increases the amount of phenolic compounds in legume seeds [66].

Fermented mung bean has been a common fermentation for many years due to the reduction in non-nutritional compounds in mung beans and the increase in nutritional value through fermentation. The antidiabetic and antioxidant properties of fermented mung bean are attributed to its free amino acids [67]. Furthermore, they have been shown to have potential chemoprophylaxis, lipid-lowering, anti-stress, hepatoprotective, and anti-inflammatory effects as a result of these health benefits. Fermented mung beans with LAB have recently emerged as a new functional food with numerous health benefits [68].

**Table 4 foods-11-03155-t004:** Examples of alkali-fermented MG foods used in different regions of the world.

Name	Grain	Microorganism (s)	Country/Region	Type of Product	Reference
Bhallae	Black gram	*Bacillus,* *Candida,* *Cryptococcus,* *Debaryomyces,* *Geotrichum,* *Hansenula,* *Kluyveromyces,* *Lactobacillus,* *Leuconostoc,* *Pichia,* *Saccharomyces,* *Enterococcus,* *Trichosporon,* *Wingea.*	India	Mild acidic, side dish	[21]
Iru	Locust bean	*Bacillus,* *Lysinibacittus,* *Staphlyococcus.*	Nigeria, Benin	Alkaline, sticky	[19]
Kawal	Leaves of legume (Cassia sp.)	*Bacillus,* *Propionibacterium,* *Lactobacillus,* *Staphlococcus,* Yeasts.	Sudan	Alkaline, strong-flavored, dried balls	[19]
Kinda	Locust bean	*Lysininbacillus.*	Sierra Leone	Alkaline, sticky	[19]
Maseura	Black gram	*Bacillus,* *Pediococcus,* *Enterococcus,* *Lactobacillus,* *Saccharomyces,* *Pichia,* *Candida.*	Nepal, India	Dry, ball-like, brittle condiment	[19]
Papadam	Black gram	*Saccharomyces.*	India	Breakfast or snack food	[19,21]
Soumbala	Locust bean	*Bacillus,* *Lysininbacillus,* *Peanibacillus,* *Brevibacillus.*	Burkina Faso	Alkaline, sticky	[19]
Wari	Black gram	*Candida,* *Cryptococcus,* *Debaryomyces,* *Geotrichum,* *Hansenula,* *Kluyveromyces,* *Sacchatomyces,* *Rhizamoeba,* *Enterococcus,* *Trichosporon.*	India	Ball-like, brittle side dish	[19]
Miso	Wheat/rice, barley, and soybeans	*Aspergillus,* *Streptococcus,* *Pediococcus,* *Micrococcus,* *Bacillus,* *Saccharomyces.*	Japan, China, U.S.	Seasoning	[19]
Shoyu (soy sauce)	Wheat and soybeans	*Aspergillus,* *Saccharomyces,* *Pediococcus,* *Lactobacillus.*	Japan, China, U.S.	Seasoning	[19]

### 2.5. Acetic-Fermented MG Vinegar

Another important bacterium that causes food fermentation are AAB species. When oxygen is in excess, AAB are able to convert ethanol into acetic acid. AAB are obligate aerobic microorganisms capable of oxidizing sugars, sugar alcohols, and ethanol. Acetic acid is the main byproduct of their metabolism. The survival of AAB can be achieved in high acetic acid concentrations and low pH [69]. Many other food-grade and industrially important microorganisms have received more attention than AAB [70]. AAB are best known for their use in the manufacture of vinegar, vitamin C, and cellulose. In the aerobic fermentation process, *Acetobaccter* is the main type of AAB, whereas in the anaerobic fermentation stage, *Clostridia* is the main type of AAB.

Fermentation products led by AAB mainly include vinegar. However, AAB are often found in products dominated by lactic acid fermentation and alcohol fermentation. As far as strain function and microbial interactions are concerned, acetic acid bacteria are often not in the core flora, and their roles need to be further studied [70]. The AAB are responsible for the acidic flavor of lambic beers through the production of acetic acid in the production of lambic beer, a naturally fermented sour alcoholic beverage, in addition to the lactic acid produced by the LAB [71,72].

Zhejiang Rosy Vinegar is a traditional condiment in Southeast China, which is produced in an open environment using semi-solid-state fermentation techniques. The most abundant organic acid is acetic acid, which increases throughout the fermentation, followed by lactic acid, which decreases continuously. Amplicon sequencing analysis has revealed that the richness and diversity of the bacterial community are highest at the start and then decrease throughout the fermentation. Acetobacter (average relative abundance: 63.7%) and Lactobacillus (19.8%) are the most common bacteria [73,74]. It should be noted that vinegar is more than just an acetic acid solution; it is a complex matrix of alcohols, acids, aldehydes, ketones, and so on [75,76].

### 2.6. Alcoholic-Fermented MG Beverage

Compared with acid and alkali fermentation, alcoholic fermentation is a relatively complex process. In alcoholic-fermented foods, the herbal aroma and hot spicy taste are characterized by their distinctive features. The complex fermentation process of the intricate raw material components produces a number of aromatic compounds under the activity of microorganisms, giving the finished product its distinct aroma [77]. The product of alcoholic fermentation is ethanol, and the representative fermented products include yellow rice wine and white wine.

Yeasts are the most well-known domesticated microbes, and they play an important role in the production of fermented foods and beverages such as bread and beer [78]. *Saccharomyces* species (primarily *S. cerevisiae*) are particularly well-suited for food fermentation because they do not secrete toxic secondary metabolites, while at the same time producing high levels of alcohol and desirable flavor molecules such as esters and phenols [79]. Yeasts are single-celled fungi common in the natural environment producing and releasing volatile organic compounds (VOCs) in the fermentation environment. These volatiles often determine the sensory properties of fermented foods [80,81]. The metabolic pathways of yeast are already widely understood; notably, *Saccharomyces*
*cerevisiae* and *Saccharomyces*
*boulardii*, which have probiotic properties and are beneficial to human health, can be given to patients as dietary supplements [82]. Many sensory bioactive volatiles and non-volatile compounds, such as peptides, amino acids, vitamins, minerals, and polyphenols, can be obtained during yeast fermentation [61,83]. The following yeasts are responsible for starch breakdown in fermented MG foods and alcoholic beverages: the genus *Brettanomyces*, *Candida*, *Cryptococcus, Debaryomyces, Galactomyces, Geotrichum, Hansenul, Hanseniaspora, Hyphopichia, Issatchenkia, Kazachstania, Kluyveromyces, Metschnikowia Saccharomycodes, Saccharomycopsis, Schizosaccharides, Sporobolomyces, Torulaspora, Saccharomycodes, Saccharomycopsis, Schizosaccharides, Sporobolomyces, Torulaspora, Trichosporon, Trichosporon, Yarrowia,* and *Zygosaccharomyces* [15]. The production of enzymes and the degradation of anti-nutritive factors are the primary functions of filamentous molds in fermented foods and alcoholic beverages [19,84]. Many fermented foods and Asian alcoholic beverages contain filamentous molds, including species of *Actinomucor, Amylomyces, Aspergillus, Monascus, Mucor, Neurospora, Parcilomyces, Penicillium, Rhizopus*, and *Ustilago* [85].

Yellow rice wine (Huangjiu) is made up of foxtail millet, oat, quinoa, broomcorn millet, and rice grain. Its production process includes soaking raw materials, cooking, adding Qu (a fermentation starter), saccharification or primary fermentation, alcoholization or secondary fermentation, pressing, filtration, sterilization, aging, etc. [86]. It is worth noting that the current production of huangjiu is still based on traditional empirical science and carried out in a non-sterile environment, resulting in different batches of product quality, which greatly restricts the standardized production and market expansion of yellow rice wine [77]. In traditional white wine (Baijiu)’s production process, the raw material is evenly combined with whole grains or powder before adding hot water. Following the fermentation of the combined grains, the husks are sent to zeng to be distilled. After that, fresh white wine is matured in ceramic or stainless-steel vessels. The stored liquor is mixed after aging for several years to obtain graded products differently [87].

## 3. Biochemical Changes and Nutritional Improvement of Fermented MGs

Fermented MG foods have been shown to produce more bioactive compounds than their unprocessed counterparts [88]. The following section reviews the changes in composition and the enhancement of nutritional value of some fermented MG foods.

### 3.1. Food Shelf-Life Extension

Longer shelf life of fermented MGs is manifested by the inhibition of contamination or growth of pathogenic microorganisms by different metabolites due to the role of fermented microorganisms, thereby preventing spoilage of food [89,90]. Fermentation is considered to be the most effective method of food preservation because of the production of organic acids, alcohols, bacteriocins, etc. with antibacterial properties under the action of microorganisms [91]. The ability of fermentation to reduce the amount of aflatoxins in MG-based meals has been recognized [92,93].

Organic acids including lactic acid, formic acid, benzoic acid, propionate, citric acid, and sobic acid are among these metabolites. These organic acids lower the starting pH and make the environment for food more acidic. By reducing the pH of the intracellular environment, these acids hinder the fundamental molecular processes of microbes [94], which lengthens the shelf life of fermented items [95]. In addition, ethanol and hydrogen peroxide are also strong inhibitors of microbial growth, and other secondary metabolites produced by LAB and some yeasts may also be antibacterial compounds, such as antimicrobial peptides and bacteriocins. These metabolites play important roles in controlling fungal growth in food and the production of mycotoxins, which are important for MG-based products, and public health concerns have arisen due to exposure to potential adverse effects on human health [96].

Some probiotic bacteria have the capacity to inhibit or stop the growth of common contaminating bacteria, which makes them a beneficial tool for producing products with longer shelf lives and higher levels of security [97]. The antagonistic systems secreted by starter cultures are thought to be responsible for the ability of LAB to suppress pathogenic strains, and *Lactobacillus casei* LUHS210, which is used in the processing of rice, soy, almond, coconut, and oat drinks, has been shown to inhibit a variety of pathogenic and opportunistic bacteria [98]. Additionally, some of the microbial starters release enzymes during fermentation that may convert mycotoxins to harmless molecules [99].

Bacteriocin production is regarded as a probiotic characteristic. It is thought that bacteria develop these peptides and, in the case of bacteriolysins, proteins to outperform other taxa, usually those that occupy the same ecological niche [100].

### 3.2. Protein and Carbohydrate Digestibility

Dietary properties of a protein depend on its structure, the presence of protein-bound antinutritional factors (protease inhibitors, phytase), and other elements including pH, temperature, and ionic strength that are directly correlated with proteolytic activity. These factors and parameters may be influenced by fermentation, contributing to increased plant protein digestibility. The pH decrease that occurs during fermentation may well encourage peptidase enzyme activity and improve protein solubility [101]. In addition, the improvement in protein digestibility has been achieved by reducing anti-nutritional factors [102].

By lowering levels of non-nutritive substances that inhibit digestive enzymes (such as trypsin and chymotrypsin inhibitors) and encouraging protein crosslinking (such as phenolic and tannin compounds), as well as through the production of microbial proteases, which partially degrade and release some matrix proteins, fermentation may improve the protein digestibility of pulses [103].

Montemurro et al. [104] investigated how germination and fermentation affected yeast’s functional and process features as well as its nutritional value. In general, fermentation-produced face cell protein has a high level of digestibility, and less starch is used. Similarly, LAB increased the starch and protein digestibility of fermented sorghum powder in vitro [105]. Natural fermentation increases the protein and starch digestibility of complementary products, which are made from sorghum, millet, and pumpkin mixed with flour and amaranth seeds in varying proportions and intended to supplement breast milk or infant formula [106].

Acidification during lactic acid fermentation has been reported to reduce glycemic index (GI) by converting fast-digestible starch (RDS) into slow-digestible starches (SDS) [107]. The nutritious qualities of pastes could be improved by adding 20 g/100 g of quinoa flour to semolina while maintaining technological and sensory excellence. Quinoa’s beneficial effects have been boosted even more by LAB during fermentation. When compared to ordinary pasta, pasta made with fermented quinoa flour had a higher nutritional profile, featuring better protein digestibility and quality, higher nutritional ratings, a lower anticipated glycemic index, and higher antioxidant potential [108].

The protein content in pearl millet grains increased significantly by 15.32% over 16 h of fermentation. Most likely, the solubilization of insoluble protein during fermentation is what caused the protein concentration to increase [109].

### 3.3. Antioxidant Activity

According to emerging research, fermentation significantly increases the nutritional value of finished products by increasing the bioaccessibility and bioavailability of bioactive components [110]. The ways in which fermentation affects the antioxidant activity of grains and foods made from fermented grains have been extensively studied. Generally speaking, depending on the raw material, the fermenting agent, and the processing circumstances, fermentation increases the antioxidant activity of fermented foods through the formation of several molecules [111]. Polyphenolic compounds, gamma-amino butyric acid (GABA), bioactive peptide, and short-chain fatty acids (SCFAs) are well-known bioactive substances that are generated and made accessible during fermentation [112].

The bioconversion of phenolic compounds from their conjugated forms to their free forms during MG fermentation allows for an increase in the total concentration of phenolic compounds. The structural breakdown of grain cell walls is facilitated by the hydrolytic activity of enzymes produced by fermenting microorganisms, which increases the bio-accessibility and bio-availability of bound and conjugated phenolic compounds [113]. The increase in polyphenol content in fermented MGs has been the subject of several studies. Quinoa and buckwheat fermented with *Pediococcus pentosaceus* and *Lactobacillus paracasei* and wheat germ, barley, rye, and buckwheat fermented with *Lactobacillus rhamnosus* and *Saccharomyces cerevisiae* both showed an increase in total phenolic content [114,115]. It has been demonstrated that *lactobacillus* fermentation increases the phenolic, avenanthramide, and flavonoid content of oats and increases the antioxidants’ capacity to prevent DNA damage in oats [116]. The overall phenolic content and antioxidant activity of the samples were significantly higher when made with quinoa sourdough in addition to locally fermented LAB [117]. A brief review on the effects of fermentation on phenolic components and antioxidant activity of whole grain cereals was published by Adebo et al. [118] This study examines how fermentation affects phenolic compounds and antioxidant activities, with the majority of available studies pointing to an increase in these beneficial components for human health. These increases are primarily caused by the breakdown of the cereal cell wall and subsequent enzyme actions that allow bound phenolic chemicals to be released and increase antioxidant activity.

GABA is produced through fermentation employing certain bacteria. To create GABA, LAB starters such *Lactobacillus plantarum*, *L. brevis*, and *Streptococcus thermophilus* are added to fermented MG foods and drinks. GABA production was impacted by temperature, pH, substrate, culture duration, and L-glutamate level during the fermentation process [119]. Oats fermented by fungi that is generally recognized as safe (GRAS) can be recommended as tempeh-like functional foods with higher GABA, more natural antioxidants, and lower phytate compared with native oats. The antioxidant activities of fermented oats were also significantly enhanced after 72 h fermentation (*p* < 0.05) [120].

Small organic monocarboxylic acids, or SCFAs, have chains that range in length from two to six carbon atoms. They are common byproducts of the fermentation of dietary polysaccharides, such as fiber and resistant starch, produced by the gut bacteria. The three SCFAs with the highest molecular weights—acetic acid, propionic acid, and butyric acid—are the most representative [121].

### 3.4. Vitamin Content

In general, in the natural fermentation of MGs, the availability of B-group vitamins may be increased, and some amino acids may be produced [33]. Even though cereals contain some vitamins, adding LAB or yeast to the MG fermentation process might boost vitamin levels [95]. Microorganisms can be employed as starter or additional cultures to boost natural fermentation products and improve nutrition and quality in a targeted manner [122]. The ability of LAB to create vitamins is strain-specific. For instance, adding *Lactococcus lactis* N8 and *Saccharomyces boulardii* SAA655 improves the concentration of riboflavin and folic acid in idli batter by 40–90% [123]. Although finger millet is mostly used to manufacture flour, its processing (germination and fermentation) increases the iron content of the grain [124].

### 3.5. FODMAP Reduction

Additionally, it has been demonstrated that fermentation is essential for lowering FODMAPs. The term “Fermentable Oligo-, Di-, and Mono-saccharides and Polyols” is abbreviated as FODMAPs. In the large intestine, microorganisms quickly digest these permeable small molecules after they have been weakly absorbed in the small intestine [125]. Patients with irritable bowel syndrome (IBS) who consume FODMAP may experience abdominal symptoms. A low-FODMAP diet, or a diet low in oligosaccharides, disaccharides, monosaccharides, and polyols, is linked to a reduction in gastrointestinal symptoms in 50–80% of patients [126].

During the fermentation process, LAB, yeasts, and fungi entirely or partially breakdown FODMAPs. The amount of these sugars in cereal foods has been the subject of numerous published investigations, but only a small number of studies have specifically examined grain-based products with low FODMAP contents [127]. The FODMAP content of whole wheat has been lowered by up to 90% by adding *Kluyveromyces marxianus* and *Saccharomyces cerevisiae* to the dough. The outcomes amply illustrate the potential of several yeast strains, particularly *Saccharomyces cerevisiae* and *Torulopsis delbrueckii*, isolated from conventional sourdough, to significantly lower fructan content during fermentation and FODMAPs overall [128].

### 3.6. Phytic Acid Degradation

The optimal pH conditions for the enzymatic breakdown of phytate, which is found in cereals as complexes with polyvalent cations such iron, zinc, calcium, magnesium, and proteins, are also provided by fermentation [29]. The amount of soluble iron, zinc, and calcium may increase dramatically with a decrease in phytate [129]. Whole grain barley and oats can have their phytic acid concentration reduced via *Rhizopus oligosporus* fermentation, and the choice of fermentation strains is a key factor in phytic acid degradation [33,130]. Additionally, the procedure of creating various breads (bulk and rye sourdough fermentation) and the type of flour used affect the degradation of phytic acid. It was discovered that both phytic acid levels and mineral bioavailability were impacted by fermenting techniques. Due to its lower pH than whole-wheat fermented bread, rye fermented by yeast dramatically decreased the quantities of phytic acid [43,131].

Haq et al. [132] reported that fermentation increased in vitro protein digestibility with a concomitant reduction in total polyphenols and phytic acid content. Different grains, including maize, sorghum, and finger millet, were fermented by LAB to minimize the amount of anti-nutrients such as phytic acid and tannins, improving the availability of protein and minerals. The effects of fermentation and certain wholemeal flours from rye, oats, and wheat were examined in order to assess factors influencing phytate levels. In freshly ground wholegrain flour dough, phytate was reduced over the course of the fermentation process, with the duration of fermentation being the most crucial element. The rate of phytate reduction was shown to be mostly unaffected by fermentation temperature [133].

## 4. Probiotic Potential of Fermented MG Foods

The probiotic potential of fermented MG foods is due in part to the probiotics that facilitate fermentation, which are often derived from the microbes that feed the fermented foods [134]. It is also partly due to the fact that beneficial metabolites or raw materials of grains are broken down into small compounds after fermentation [135]. Although some fermented MG foods contain beneficial bacteria, such as LAB and yeast, during the fermentation process, these microorganisms may be destroyed during the heating process, and the finished fermented foods do not contain live microorganisms. However, after heating or pasteurization, the fermented foods are also rich in small compounds broken down by the raw materials themselves or metabolic products of microorganisms during the fermentation process. [40]. There is proof that some foods have health benefits that go beyond the actual dietary components. Healthy fermented foods may benefit from nutritional changes in their components, immune system modulation, the existence of bioactive substances that alter gut and systemic function, or the regulation of gut microbiota composition and activity [136].

### 4.1. Important Source of Probiotic Microorganisms

Fermented MG foods are well-liked all over the world due to their high nutritional content as well as the abundance of beneficial probiotics they contain. Live microorganisms that improve the host’s health when given in sufficient quantities are referred to as probiotics [137]. The ways in which probiotics improve the health characteristics of the mechanism are not completely clear, but usually we think of interference with pathogens, rejection or antagonism, immunomodulatory activity, anticancer and anti-mutation activity, the alleviation of lactose intolerance symptoms, lowering of serum cholesterol levels, lowering of blood pressure, prevention of and reduction in the incidence of diarrhea and duration, prevention of bacterial vaginal disease and urinary tract infection, maintenance of mucosal integrity, and improvement of periodontal quality [138,139].

Currently, researchers focusing on the human microbiome recommend using fermented foods in healthy diets to temporarily boost the number of live bacteria in our gut [51]. Not coincidentally, most probiotic strains are derived from the fermentation microbiome [140]. Isolates from foods with probiotic potential are used as adjuvant cultures in foods to enhance the palatable sensory properties and ensure microbiologically safe food [141].

A probiotic must survive gastrointestinal digestion and adhere to intestinal epithelia in order to be beneficial. The hydrophobicity and auto-aggregation capacity of probiotic microorganisms influence adherence [142]. Probiotic dosage, duration, and carrier matrix may all have an impact on how long probiotics survive and stay in the colon.

One of the biggest and fastest-growing markets is for probiotic products that improve intestinal health [143]. According to the definition given by probiotics, they are “live microorganisms that when are administered in an adequate amount confer a beneficial effect on the host health” [11]. Probiotics can be taken as dietary supplements or as a component of fermented MG foods. The most widely used genera of probiotics that may be found on the food market are the *Lactobacillus* and *Bifidobacterium* [144,145]. As their ingestion poses no dangers to the host’s health, these species have generally been accorded a generally recognized as safe (GRAS) status and a qualified presumption of safety status [37].

Despite the fact that some fermentation systems only have a few dominating groups, strain variations and population dynamics can be highly intricate [146]. The combination of cereals used, the region and/or country, and other factors can all affect the microbiota in fermented MG foods [147].

Recent research suggests that giving infants and kids an antibiotic while they take probiotics can lower their risk of developing acute respiratory and gastrointestinal infections. Probiotic supplements have been found in studies to treat various infections more successfully than a placebo, which reduces the need for antibiotics [148]. Evidence suggests that the therapeutic impact of the probiotic *Lactobacilli* mixture chosen was linked to the generation of transferable tolerogenic T cells in the peripheral, central nervous system and mesenteric lymph nodes. These findings provide us with a better knowledge of the host–commensal interactions that lead to therapeutic effects in autoimmune illnesses and suggest a therapeutic potential of oral administration of a mixture of probiotics [149].

### 4.2. Enhancement of Anti-Inflammatory and Immunomodulatory Ability

Some fermented MG foods contain antioxidants that work by neutralizing free radicals and controlling the activity of antioxidant enzymes, reducing oxidative stress, improving the amelioration of inflammatory responses, enhancing the adaptive capacity of the immune system, etc. As a result, fermented MG products can defend against chronic inflammatory diseases, which are known to be the leading cause of mortality worldwide [150].

According to research so far, adding fermented foods to one’s diet greatly lowered the health issues linked with diabetes by boosting the host body’s antioxidant and anti-inflammatory defenses. These studies, however, imply that not all fermented foods have antidiabetic qualities. The bioactivity of the final product is significantly influenced by a number of variables, including the fermentation process, the microbial strain utilized in the fermentation, and the raw ingredients used [151].

It has been demonstrated that the fermentation of traditional dietary fiber-rich ingredients, including soybean germ, wheat germ, or rice bran made using conventional fermentation methods, produces novel bioactive compounds with advantageous immune, hypoglycemic, and anti-inflammatory properties [25]. The presence of phenolic aldehyde in fermented rice bran has been recognized, and there is particular experimental support for the bioactive chemicals’ advantageous effects on mental health. In contrast to a control group, the oral dose of fermented rice bran extract lessened tiredness and stress [152]. It is known that fermentation of traditional foods such as mung beans, buckwheat sprouts, and lentils considerably increases the quantity of GABA that is readily available.

The three primary isoflavones with the strongest anti-inflammatory effects that are present in fermented soybeans are genistein, daidzein, and glycitein [153]. Recent studies have demonstrated that isoflavones are the primary mediators of systemic allergic inflammation in basophils, mast cells, and eosinophils [154]. There have been some reports of a decreased prevalence of Clostridium leptum in inflammatory illnesses such as inflammatory bowel disease (IBD) [155]. Additionally, *Bifidobacterium and Bacteroides fragilis* were more prevalent in fermented soybean [156]. According to a study, consumption of high-fiber rye products may change the gut flora of Chinese people who have *Helicobacter pylor* infection. Such effects coincide with positive changes in the content of SCFAs and are linked to glycemic characteristics that are impaired [157].

Traditional antioxidant and anti-inflammatory diets high in antioxidants may provide some protection against antidepressants; however, it is unclear to what extent fermented MG foods contribute to mental health [158]. There appears to be a gut “inflammatory microbiota” that may affect mood through intestinal permeability, systemic lipopolysaccharide (LPS) burden, or even direct communication with microbes in the brain. Western eating habits contribute, at least in part, to the formation of this inflammatory microbiome. Studies have shown that people with high levels of depression, anxiety, and chronic depression typically consume foods high in fat or sugar and low in nutritional value, which increases the likelihood of aggregation of inflammatory microbiota [159,160]. According to preliminary research in rodents, it is possible to treat the behavioral effects of an inflammatory microbiome by giving patients healthy bacteria [26]. Immunobiotic strains improved resistance to rotavirus infections and reduced the disruption of intestinal homeostasis brought on by intraepithelial lymphocytes. These effects were achieved by modulating the toll-like receptor 3-triggered immune response in intestinal epithelial cells [161].

### 4.3. Maintaining the Balance of Gut Microbiota

The intestinal microbiota is a complex and diverse collection of microorganisms composed of 400–500 different species of metabolically active bacteria that have a strong impact on the intestinal function and the health of the host, with the majority of bacteria belonging to the phyla *Firmicutes* and *Bacteroidetes*. Their effects on the control of energy metabolism and the development of the immune system have a significant impact on the host’s health [162]. Live probiotics are administered to maintain the balance of the gut microbiota and improve intestinal health in general. As a result, people are increasingly requesting and consuming fermented MG foods [163].

Understanding the microbiome gut–brain axis and its significance in the treatment of neurodegenerative illnesses has advanced significantly over the past ten years. Scientists have recently demonstrated how the gut lumen communicates quickly with the brain after meals and the significance of the microbiota in controlling homeostasis and appropriate signaling with the finding of “neuropod cells” [164]. In their study of the effects of two different oat samples on the gut flora, Kedia et al. [165] showed that *Lactobacilli* and *Bifidobacteria* might flourish in a fecal mixed culture. These fractions also allowed for the growth of enterobacteria, which are regarded as being advantageous for the gut. Another study found a positive correlation between maternal consumption of fermented foods and sleep duration in one-year-old infants, and we hypothesized that this relationship may be the result of changes in the gut microbiota of expectant mothers after consuming fermented foods [166].

Recent research reveals that flavonoids’ health-promoting effects are directed at the metagenome of human gut bacteria and have an evolutionary basis. Flavonoids may control gut microbial metabolism, according to functional research employing homologous groups of bacterial target proteins [167]. According to experimental research, common dietary polyphenols that are fermented produce biotransforming phytochemicals that are better able to trigger favorable changes that promote microbial development [168].

The relationship between fermented dairy products and the development of advantageous gut flora has been extensively discussed. The gut microbiota can be positively impacted by (non-dairy) fermented foods and herbs, which is significant since it may have an impact on long-term gut–brain communication [169,170]. For instance, it has been demonstrated that the traditional sugar isomaltose, which is included in foods such as honey, sake, miso, and soy sauce, has positive effects on the proliferation of the bacteria bifidobacterium and lactobacillus in both animals and people [171,172].

SCFAs that are beneficial to the gastrointestinal tract include butyrate, acetate, and propionate. SCFAs, which are produced by probiotics through the fermentation of indigestible carbohydrates, improve the intestinal barrier, inhibit pathogen development and the production of toxic elements, and are grown by intestinal cells as colonic cells [173]. As a result, consuming fermented MG foods and beverages with probiotic fermentation can help to improve glucose metabolism and prevent the development of type 2 diabetes. These novel fermented foods and beverages could serve as carriers of probiotics, prebiotics, and/or bioactive compounds to help prevent metabolic changes in diabetic pathology. More research is needed, however, on microbial metabolites, which are also important and have implications for gut health.

## 5. Fermentation of Grain Mixtures Is a Better Strategy

When combined in a specific ratio, the complex nutrient makeup of several grains can significantly alter the qualities of food. The availability of nutrients for fermentation and the growth and metabolism of the microorganisms may fluctuate in response to two substrates. Mixed culture fermentations, on the other hand, offer complicated development patterns that can also significantly impact the organoleptic and functional qualities of the meal.

Mixed fermentation of cereals and legumes is a better strategy. Cereals include little in the way of B-group vitamins and minerals such as iron and zinc. Whereas cereals are low in lysine but high in cysteine and methionine, legumes are high in lysine but low in these other amino acids. Therefore, based on the composition of the amino acids, the overall protein quality of such a mixture is superior to that from either protein source alone [10]. However, it is known that the presence of various antinutrients, including protease and amylase inhibitors, lectins, flatus factors, metal chelates, antivitamins, goitrogens, cyanogens, tannins, toxic phenolic glycosides, and amino acid derivatives, can impact the consumption of legumes [174]. Therefore, it is crucial to properly prepare the cereal–legume mixture before eating to reduce or remove these antinutrients. This was accomplished to co-ferment finger millet and horse gram to create a cheap, protein-rich food. Finger millet and horse gram flour mixtures at various ratios (2:1, 3:1, 4:1, and 5:1) underwent natural fermentation for 24 h. According to biochemical examination, pH (6.6–4.2) and starch content (25.52%) decreased only moderately, but titratable acidity (0.168–1.046%), soluble proteins (1.1 times), and free amino acids (2.6 fold) significantly increased at 16 h. The newly developed product may be utilized to address the issues with protein and energy deficiency [175].

Mixed fermentation of major grains and minor grains is a better strategy. Using several LAB strains, Rathore et al. [176] compared single- and mixed-cereal (malt, barley, and barley combined with malt) beverages. According to the study’s findings, mixed cereal substrates produced less organic acid than single-cereal beverages. The microbial populations, however, were comparable across all substrates. Future research on sensory characteristics and consumer acceptance will find great interest in these findings. However, malt proved the most effective growing medium for microbes when used to make single or mixed drinks. In order to produce mixed beverages, Freire et al. [177] fermented rice and maize using *Lactobacillus acidophilus* LACA 4 and *L. pantarum* CCMA 0743 for 36 h at 37 °C. Lactic and acetic acid were the two principal organic acids. The beverages’ high scores in comparison to unfermented beverages and positive sensory acceptability showed the marketability of these drinks. Oat bran (20 g/100 g) added to a whole-wheat flour baking formula resulted in loaves containing 1.3–1.4 g of β-glucan per serving, enough to meet the daily recommended allowance. High-acidity bread known as sour-dough bread demonstrated a significant benefit when oat bran was added to the recipe. Compared to yeast-fermented breads, this form of bread had a lesser fall in β-glucan and viscosity [178].

Mixed fermentation of grains and milk is a better strategy. Few attempts have been undertaken to create probiotic foods using alternative fermentation substrates; the majority of probiotic foods available on the market today are milk-based. For a substantial portion of the population, particularly in underdeveloped nations, cereals represent the most affordable source of protein and calories. Cereals are a rich source of fiber, but milk is insufficient. Whereas cereal proteins have low digestibility and are deficient in some amino acids, milk proteins have significant biological value and good amino acid profiles. Better nutrition is provided by cereals and milk, which will ultimately result in functional products with added value [179,180].

## 6. Conclusions

New functional foods with positive health effects are in high demand due to changing eating patterns brought on by urbanization and rising consumer knowledge of good diets. People who have lactose intolerance, milk allergies, or who follow a low-lipid or vegan dietary pattern can consume fermented MG products. Additionally, they are regarded as cutting-edge probiotic delivery systems for probiotics and prospective functional foods. However, further in vivo research, such as human clinical trials addressing matrix combinations and dosages in diverse populations, is required to support the health advantages of consuming fermented non-dairy beverages. There are still many obstacles ahead, and selecting the right probiotic strain to employ in fermented foods is critical. Food fermentations, according to the literature reviewed for this study, are an effective way to increase the nutritional content of legume- and cereal-based meals in terms of dietary protein and micronutrients. As a result, the development of food fermentations can help to promote the goals of sustainable development by increasing the diversification of protein sources and the accessibility of nutritious and nutritionally well-balanced stable meals suited for a wide range of consumer groups globally. The industrialization of traditional procedures that allow better control over the manufacture of fermented goods suitable to global customers is a significant problem.

## Data Availability

Not applicable.

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
