# Peer review of "Fermented Minor Grain Foods: Classification, Functional Components, and Probiotic Potential"

_foods, 2022, doi:10.3390/foods11203155_

Round 1

Reviewer 1 Report

The manuscript tried to review the fermented minor grain in some aspects including the classification, biochemical and nutritional changes, and potential probiotics and proposed a new strategy to improve the food products. The authors have tried to write the manuscript in English, but there are too many grammatical errors that need high attention. Please improve the manuscript to an appropriate English editing service.

Review of Foods-1928595

General

The manuscript tried to review the fermented minor grain in some aspects including the classification, biochemical and nutritional changes, and potential probiotics and proposed a new strategy to improve the food products. The authors have tried to write the manuscript in English, but there are too many grammatical errors that need high attention. Please improve the manuscript to an appropriate English editing service.

Abstract:

Please imply a brief recommendation for further research and development in the fermented MGs to fill the gap in the given works of literature.

Line 10             : “Fermented minor grain(MG)….” Please put space after grain.

Line 16             : “Specific discussion will focus…” Please change to past tense as the review has been done.

Introduction:

There are many grammatical errors and words misused, here are some examples:

Line 44-45       :“ Furthermore, the role of fermented foods in the human diet is discussed.” Please describes what is discussed as this sentence is not clear.

Line 58-59       :” Fermented MG foods come from a wide range of sources and are easy to operate and control...” What does “easy to operate and control” refer to? Please improve the sentence.

Line 65-66       :” For example, in Africa and South America, cereals like sorghum, maize porridges have been a staple there.” This sentence is not clear what do the authors refer to? Please clarify.

Line 68-70       :” Traditional fermented MG foods are important to Africans not only for …… ingredients.” Please improve the sentences

Line 96             : “artisanally” what does it mean? This is not a proper word for scientific writing.

Main body

The authors have described the fermented MG foods classification and biochemical changes and nutritional improvement, however, there are no physical changes such as textural properties, viscosity properties, and other functional parameters. Please add this review in a separate section.

Line 119           : ” autochthonous” again, is not proper word.

Line 132           : “Virtually” Please improve the vocabulary.

Line 195           : Table 1. “snake” ???

Line 201           :”… yeast is also present, and will produce amount of alcohol…” please change “will” to “may”

Line 298           : “Furthermore, they have been shown…” supposed to be: “They have shown…”

Line 375           : “Fermented MG foods have been shown...” see the previous comment.

And so on.

Reviewer 2 Report

This review article focusses on different aspects of fermented minor grain foods starting from classification according to fermentation process, followed by nutritional improvement and probiotic potential caused by the fermentation process. The content of the paper is suitable for Special Issue: Functional Ingredients in Minor Grain Crops.

Page 1, Lines 32-33: „MGs have higher nutritional value than stable grains, which are frequently high in dietary fiber, vitamins and minerals (especially trace elements like iron and zinc) [5].“

Please clarify the above sentence. It’s not quite clear what authors wanted to say. Who is frequently high in dietary fiber, vitamins and mineral? From current sentence it appears that MG grains are nutritionally superior compared to staple (also change stable to staple) grains which are richer in nutritionally valuable components such are fibers, vitamins, and minerals.?

Page 3, Line 136: Erase the full stop in the sentence following the word Asia “… found in Africa and Asia. and the …”

Tables 1 to 4: I suggest that the name of the column Cereal in tables 1-4 be replaced with Grain. Although most grains listed in this column are cereals, some of them are not. I think that Grain is more precise and in line with the title of the article itself.

Page 5, Lines 204-205: „The fermentation time of acid fermented food is often not long, 1-3 d can be completed fermentation, and pH is low, during 3.5-4.4.“

I found this sentence a little bit confusing. Put “day” instead of just “d” and what is meant by “during 3.5-4.4”. Is this the final pH of the product? Please rewrite this sentence to avoid any misunderstandings.

Page 12, Line 401: Please elaborate or indicate which press cakes are processed by Lactobacillus casei LUHs210. Actually, press cake is a very broad term, it could be a filter cake formed in a filter press or oil cake after pressing or extraction.

Page 13, Line 426: Put the full stop in the sentence following the reference number [105] “… in vitro [105].”

Page 14, Lines 485-486: “In general, cereals natural fermentation lowers their carbohydrate content as well as several non-digestible poly and oligosaccharides.”

It is not quite clear how is this sentence (statement) connected to the rest of the text in part 3.4 Vitamin content. Whether the increase in the availability of vitamin B is a consequence of the decrease in the content of carbohydrates? Please elaborate this more thorough.

Pages 15-16, Lines 555-561: I found this sentence rather confusing. Authors should consider to rewrite it.

Pages 16, Lines 565-566: “Probiotics can control the need for antibiotics in kids, according to recent studies, which can prevent the development of antibiotic resistance.”

I found this sentence confusing – it should be rewritten. What is meant by “need for antibiotics in kids”? I presume that it is the need for children treatment with antibiotics. If that is a case and probiotics prevent the development of antibiotic resistance that implies that they favor the effect of antibiotics. Maybe I am missing some more subtle effect here. 

Pages 17, Lines 627-629: “The traditional antioxidant and anti-inflammatory diets rich in antioxidants may provide some degree of protection against antidepressants, so to what extent fermented MG foods contribute to mental health [159].”

I am having trouble to understand this sentence, especially the second part of it “so to what extent fermented MG foods contribute to mental health”. Please rewrite it.

Page 17, Line 633: Put a capital letter in word Studies (start of a new sentence): “ …microbiome. Studies …”

Literature   

-        The complete reference list should be carefully checked. It appears that some information’s are missing in few references such as lack of page numbers or the end page numbers, etc.

-       In the references from number 35 (line 835) to 47 (line 857) some numbers appear before the name of the first author, which are not the number of the reference in order. This needs to be corrected. Below is the list of these references:

35. 41.Holzapfel, W. H. Appropriate starter culture technologies for small-scale fermentation in developing countries. Int J Food 835 Microbiol 2002, 75(3), 197-212. 836

36. 43.Oyewole, O. B. Lactic fermented foods in africa and their benefits. Food Control 1997, 8(5-6), 289-297. 837

37. 44.Wuyts, S.; Van Beeck, W.; Allonsius, C. N.; van den Broek, M. F. L.; Lebeer, S. Applications of plant-based fermented foods 838 and their microbes. Curr Opin Biotechno. 2020, 61, 45-52. 839

38. 35.Lee, C. H. Lactic acid fermented foods and their benefits in asia. Food Control 1997, 8(5-6), 259-269. 840

39. 36.Chilton, S.; Burton, J.; Reid, G. Inclusion of Fermented Foods in Food Guides around the World. Nutrients 2015, 7(1), 841 390-404. 842

40. 45.Xu, Y.; Zhou, T.; Tang, H.; Li, X.; Chen, Y.; Zhang, L.; Zhang, J. Probiotic potential and amylolytic properties of lactic acid 843 bacteria isolated from Chinese fermented cereal foods. Food Control 2020, 111, 107057. 844

41. 46.Caplice, E. ; Fitzgerald, G. F. Food fermentations: role of microorganisms in food production and preservation. Int J Food 845 Microbiol 1999, 50(1-2), 131-149. 846

42. 37.Adesulu-Dahunsi, A. T.; Jeyaram, K.; Sanni, A. I. Probiotic and technological properties of exopolysaccharide producing 847 lactic acid bacteria isolated from cereal-based nigerian fermented food products. Food Control 2018, 92, 225-231. 848

43. 38.wuoha, C. I. ; Eke, O. S. Nigerian indigenous fermented foods: their traditional process operation, inherent problems, im-849 provements and current status. Food Res Int 1996, 29(5-6), 527-540. 850

44. 47.Pasolli, E.; De Filippis, F.; Mauriello, I. E.; Cumbo, F.; Walsh, A. M.; Leech, J.; Cotter, P. D.; Segata, N.; Ercolini, D. 851 Large-scale genome-wide analysis links lactic acid bacteria from food with the gut microbiome. Nat Commun 2020, 11(1). 852

45. 39.Gupta, V.; Pakwan, C.; Chitov, T.; Chantawannakul, P.; Manasam, M.; Bovonsombut, S.; Disayathanoowat, T. Bacterial 853 compositions of indigenous Lanna (Northern Thai) fermented foods and their potential functional properties. PloS One 2020, 854 15(11), e0242560. 855

46. 40.Adebo, O. A. African Sorghum-Based Fermented Foods: Past, Current and Future Prospects. Nutrients 2020, 12(4), 1111. 856

47. 42.Petrova, P. ; Petrov, K. Lactic Acid Fermentation of Cereals and Pseudocereals: Ancient Nutritional Biotechnologies with 857 Modern Applications. Nutrients 2020, 12(4), 1118.

Round 2

Reviewer 1 Report

The authors have successfully answered all reviewers' questions and improved the manuscript and the manuscript can be considered to be accepted for publication in Foods.